# Ultrafast photoluminescence and multiscale light amplification in nanoplasmonic cavity glass

Piotr Piotrowski [1,2] ✉, Marta Buza[3], Rafał Nowaczyński[2,4], Nuttawut Kongsuwan[5,6], Hańcza B. Surma[1,3], Paweł Osewski[3], Marcin Gajc[3], Adam Strzep[7], Witold Ryba-Romanowski[7], Ortwin Hess [8] ✉ & Dorota A. Pawlak [1,2,3] ✉

Interactions between plasmons and exciton nanoemitters in plexcitonic systems lead to fast and intense luminescence, desirable in optoelectonic devices, ultrafast optical switches and quantum information science. While luminescence enhancement through exciton-plasmon coupling has thus far been mostly demonstrated in micro- and nanoscale structures, analogous demonstrations in bulk materials have been largely neglected. Here we present a bulk nanocomposite glass doped with cadmium telluride quantum dots (CdTe QDs) and silver nanoparticles, nAg, which act as exciton and plasmon sources, respectively. This glass exhibits ultranarrow, FWHM = 13 nm, and ultrafast, 90 ps, amplified photoluminescence (PL), $\lambda_{em} \cong 503$ nm, at room temperature under continuous-wave excitation, $\lambda_{exc} = 405$ nm. Numerical simulations confirm that the observed improvement in emission is a result of a multiscale light enhancement owing to the ensemble of QD-populated plasmonic nanocavities in the material. Power-dependent measurements indicate that >100 mW coherent light amplification occurs. These types of bulk plasmon-exciton composites could be designed comprising a plethora of components/functionalities, including emitters (QDs, rare earth and transition metal ions) and nanoplasmonic elements (Ag/Au/TCO, spherical/anisotropic/miscellaneous), to achieve targeted applications.

Recent advances in nanofabrication have enabled a novel design approach for optical devices, which has led to increased interest in enhancing nanoscale light sources using plasmonic nanoparticles (NPs)[1–4]. At resonant wavelengths, plasmons in metallic NPs can be excited, giving rise to localized surface plasmon resonance (LSPR). It is the locally increased electromagnetic fields at the metallic nanostructures, caused by the surface plasmon resonance, that can amplify the optical response of nanosystems[1]. Optical phenomena affected by LSPR include fluorescence/photoluminescence[5,6]; optical non-linearities[1,7]; and Raman scattering, including selective surface-enhanced Raman scattering[8,9]. Among these phenomena, plasmonically-enhanced luminescence seems particularly enticing due

[1]Centre of Excellence ENSEMBLE3 sp. z o.o, Wolczynska 133, Warsaw, Poland. [2]Faculty of Chemistry, University of Warsaw, Pasteura 1, Warsaw, Poland. [3](Formerly at) Institute of Electronic Materials Technology, Wolczynska 133, Warsaw, Poland. [4]Faculty of Materials Science and Engineering, Warsaw University of Technology, Woloska 141, Warsaw, Poland. [5]Quantum Technology Foundation (Thailand), 98 Soi Ari, Bangkok, Thailand. [6]Thailand Center of Excellence in Physics, Ministry of Higher Education, Science, Research and Innovation, Bangkok, Thailand. [7]Institute of Low Temperature and Structure Research PAS, Okolna 2, Wroclaw, Poland. [8]School of Physics and CRANN Institute, Trinity College Dublin, Dublin 2, Ireland. ✉e-mail: piotr.piotrowski@ensemble3.eu; ortwin.hess@tcd.ie; dorota.anna.pawlak@ensemble3.eu

to its potential use for a wide range of real-life applications. This is particularly manifested in plexcitonic systems where interactions between plasmonic nanoparticles and closely-located excitonic systems such as quantum dots or molecules are facilitated[10,11]. For instance, enhanced emission due to exciton-plasmon coupling in plexcitonics has been exploited to build a surface plasmon amplification by stimulated emission of a radiation device, called a spaser[5,12–17]. In a lasing spaser, identical spasers synchronize their dipole moment oscillations, producing coherent radiation[18,19]. Thus, they become an effective emitter of strong far-field light. Moreover, owing to their intrinsic light-matter hybridization, plexcitonic systems can facilitate controlled coupling[10] and room-temperature strong coupling[20]. These pave the way towards development and optimization of applications such as low-threshold lasers, biomedical detection techniques, and quantum information processing methods[21], allowing for e.g., room temperature quantum plasmonic immunoassay sensing[22] and quantum sensing[11,23]. Further potential applications include plasmonic transistors, modulators, and photoenergy-converting nanomaterials[11,24]. Combining plexcitonics with nonlinear optical properties can lead to optical switching, amplification, and regulation of light–matter interactions at the nanoscale[21].

Typical luminescence sources for plexcitonic systems include semiconductors[1,2,25,26] and fluorescent molecules[13,17]. The use of low-dimensional semiconductor structures such as quantum wells, nanowires, and quantum dots (QDs) is particularly appealing, as their properties can be easily engineered by manipulating their composition[27], strain[28] or doping[29]. In particular, QDs are highly efficient emitters that surpass organic molecules in terms of their chemical/ thermal stability and a high photobleaching threshold, which are essential for stable and reliable device performance[2,30]. Additionally, their size-dependent luminescence facilitates the adaptation of QD-based materials to a plethora of wavelength-dependent applications, such as display devices, light-emitting diodes, and photoconductive photodetectors[31–34]. One of the main parameters limiting the applicability of QDs is spectral resolution, defined by the full width at half maximum (FWHM) which, in the currently available systems, goes down to 20–30 nm[31,32].

To date, the exciton-plasmon coupling has mostly been studied in colloids or surface-assembled systems[2,14,32], whereas their bulk counterparts have largely been neglected. Therefore, it is urgent to fill this gap in order to exploit the potential of plexcitonic materials as optical elements, including fibers; optical waveguides; photosensitive glasses; transmitting, reflecting, and chirped Bragg gratings or phase plates[35]. Both polymers[36,37] and glasses[35,38–41] have been utilized as matrices for doping with QDs or plasmonic NPs. However, the incorporation of NPs into polymers is associated with many obstacles, such as the

agglomeration in polymer precursors and the separation from the matrix. This restricts the number of suitable polymers, complicates the fabrication process[42] and ultimately reduces the potential application of such materials. In view of the above, glass appears to be an advantageous choice for the matrix material.

For a long time, the incorporation of semiconductor nanocrystals into various types of glass was performed through the thermal development from precursors added to the matrix[43–45]. However, this resulted in the generation of QDs with an inhomogeneous size distribution, which impeded their utility in the fabrication of high-performance optical components. Furthermore, this procedure has additional drawbacks, such as difficulty in arbitrarily choosing the shape of precipitates and in introducing more than one kind of dopant. On the other hand, the recently demonstrated Nanoparticle Direct Doping (NPDD) method overcomes the aforementioned issues and makes control over the NPs added to the material possible[39]. It enables the manufacturing of bulk glass-based composites doped directly with plasmonic particles[39] or quantum dots[41] with different chemical compositions and sizes in a direct way, without employing chemical reactions to form the NPs.

Here, we demonstrate an easily manufactured, transparent material for high-efficiency emission enhanced by plasmonic effects. It is accomplished by the fabrication of volumetric glass-based plexcitonic nanocomposites simultaneously doped with cadmium telluride quantum dots (CdTe QDs) and silver nanoparticles (nAg) obtained with the NPDD method, exhibiting ultrafast and spectrally ultranarrow luminescence. The presence and distribution of QDs within the material are demonstrated with fluorescence lifetime imaging microscopy, as illustrated in Fig. 1. Finite-difference time-domain (FDTD) simulations prove that this exceptional optical response originates from the interaction between CdTe quantum emitters and the self-arranged framework of plasmonic NPs in the obtained materials. Theoretical modeling also reveals that the quantum emitters are effectively localized in nanocavities between at least two plasmonic NPs. Moreover, the power dependence of (PL) indicates coherent light amplification in the system, while temperature dependence provides additional insight into the nature of the emission. This study potentially opens up a new field of three-dimensional plasmonic scaffolds enclosed in a bulk material as well as new random-cavity-based lasers.

## Results

To obtain a bulk plexcitonic material, we prepared glass rods doped simultaneously with QDs and nAg. Moreover, we investigated the plasmon-exciton interactions to uncover the influence of plasmonic NPs on the emission properties of the QDs in the obtained volumetric materials.

Sodium borophosphate glass ($Na_5B_2P_3O_{13}$, NBP, Fig. 2a) was chosen as the matrix material, due to its transparency over a broad wavelength range and low melting point. In a low-melting matrix, metal/semiconductor NPs, which typically have higher melting temperatures, can be incorporated without the risk of destroying them during the solidification process[39]. Ag plasmonic NPs were chosen since they support surface plasmons in the visible range with relatively low optical losses. Furthermore, 1.5 nm diameter CdTe QDs were chosen as the excitonic emitters as they exhibit absorption which overlaps with the LSPR of the used nAg, 20 nm in diameter (Fig. 2b–c). Next, all raw materials in powder form were introduced into the crucible and they were molten and solidified through the directional solidification process in the NPDD method (Fig. 2d). Plexcitonic nanocomposites made of NBP glass co-doped with QDs and nAg (NBP:CdTe,nAg) were fabricated and, as a reference, nanocomposites made of NBP glass doped solely with QDs (NBP:CdTe). NBP:CdTe rods appear bright yellow (Fig. 2e). This is similar to the color of the aqueous dispersion of the same CdTe QDs, a photograph of which can be seen along with the corresponding extinction and luminescence spectra in

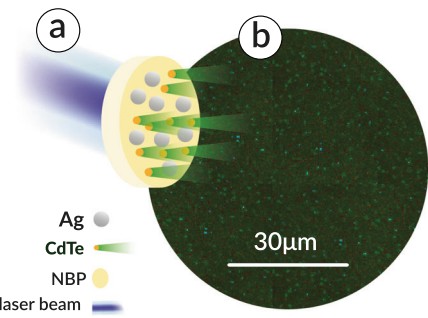

**Fig. 1 | Distribution of the quantum dots in the sodium borophosphate (NBP) glass-based nanoplasmonic composite with CdTe quantum dots and silver nanoparticles (nAg), NBP:CdTe,nAg. a** Scheme showing the generation of the optical signal in the material. **b** fluorescence-lifetime imaging microscopy showing the optical signal originating from the CdTe QDs (diameter 1.5 nm, $\lambda_{em}$ = 510 nm) in the material; luminescence is depicted as green dots.

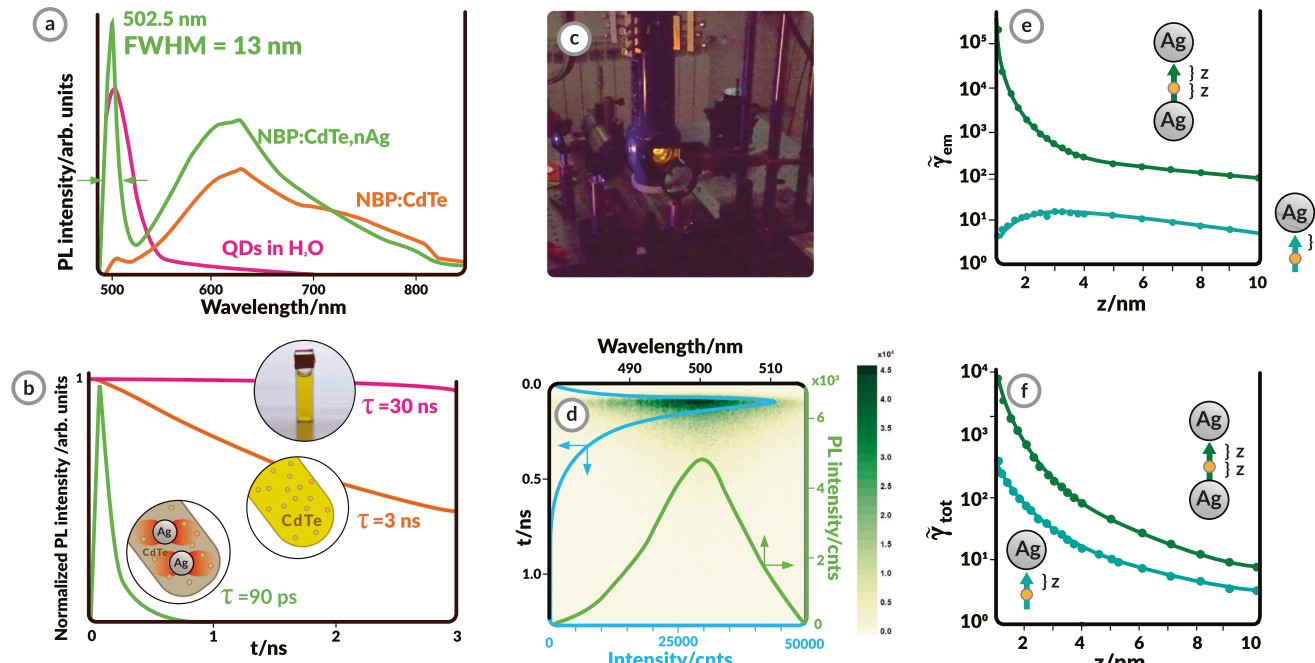

**Fig. 2 | Room-temperature ultranarrow photoluminescence (PL) and ultrashort PL lifetimes observed in the glass QDs-nAg (where QDs: quantum dots, nAg: silver nanoparticles) nanocomposites. a**, **b** PL spectra (**a**) and normalized time-resolved fluorescence (**b**) of the QDs in water (pink) and the NBP:CdTe (orange) and NBP:CdTe,nAg (green) nanocomposites. Both nanocomposites show a peak at 502.5 nm corresponding to exciton emission and a broadband PL corresponding to surface defects. The nanocomposite co-doped with QDs and nAg shows an enhanced and narrowed QD PL (full width at half maximum FWHM = 13 nm) and an almost three orders-of-magnitude decreased PL lifetime. **c** Image of the sample PL when excited with a 488 nm laser. **d** Image of the narrow-emission decay time from the streak camera with the spectrum (green line) and decay time curve (blue line). **e**, **f** Simulations (with lines to guide the eye) of the PL emission enhancement factor $\tilde{\gamma}_{em}$ (**e**) and total decay enhancement (Purcell factor) $\tilde{\gamma}_{tot}$ (**f**) of a CdTe QD situated at a distance $z$ from a single nAg particle (light blue; inset: zoomed in) and in-between two nAg particles separated by a 2$z$ distance (dark green).

Fig. 2c. The NBP:CdTe,nAg rod exhibits a different color (Fig. 2f) resulting from a different extinction spectrum due to contributions of nAg surface plasmons (Supplementary Fig. 1).

PL characterization of the fabricated materials demonstrated that the NBP:CdTe,nAg rod exhibits ultranarrow (Fig. 2a) and ultrafast (Fig. 2b) emission at room temperature, as opposed to NBP:CdTe and the CdTe QDs aqueous solution. The QDs in water exhibit an exciton emission peak with a maximum at 505 nm and a full-width-at-half-maximum (FWHM) of 34 nm. On the other hand, the slightly shifted exciton PL at 503 nm is barely noticeable in the NBP:CdTe glass rod, while a broad emission in the range from 530 to 800 nm dominates the spectrum. This broad emission, observed with the naked eye in Fig. 2c, can be assigned to the surface defects and cadmium vacancies in the CdTe QDs, based on the previous studies of bulk CdTe[46–49]. Its appearance in NBP:CdTe is attributed to the changes in the QDs' surface chemistry during the technological process and its high intensity is connected with the high surface-to-volume ratio in the QDs. What is most striking is that in the composite simultaneously doped with nAg and QDs, the defect emission decreases while the exciton spontaneous emission at 503 nm increases approximately 50 times. Additionally, the FWHM of the exciton emission is only 13 nm, which is 43% of the FWHM of CdTe QDs in water. Currently, available Cd-based QDs exhibit approximately 20–30 nm of FWHM exciton emisson. Decreasing the QD's bandwidth is crucial to increasing the efficiency of optical devices, such as LEDs, as it leads to increased color gamut and a higher luminous efficacy of radiation[31,34].

Another important effect that emerges from the exciton-plasmon interactions in the presented system is the exceptionally fast PL decay time (τ) of the NBP:CdTe,nAg composite, which exhibits $τ$ = 60–90 ps (Fig. 2b). This is three orders of magnitude faster than for the water-dispersed QDs (30 ns). Additionally, in Fig. 2d, the PL lifetime decay is

illustrated with the streak camera image, demonstrating the super-imposed decay curve and PL spectrum.

To explore and gain insight into the observed ultranarrow and ultrafast emission in the nanocomposite co-doped with exciton emitters and plasmonic NPs, finite-difference time-domain (FDTD) simulations were performed. It confirms that enhancement of the electromagnetic fields in the direct surroundings of the QDs located in nanocavities between two nAg particles plays the leading role in amplifying the PL parameters of NBP:CdTe,nAg. Indeed, an increase in intensity and decrease in PL lifetime are expected for an emitter in a suitable nanoplasmonic environment[22,50–52]. A single plasmonic nano-particle is known to quench PL emission when an emitter, such as a QD, is placed too close to the surface of a nanoparticle[3,53]. Thus, the enhancement by a single nanoparticle cannot account for the drama-tically enhanced PL intensity observed here. On the other hand, it is likely that randomly distributed CdTe QDs could be located inside an optical nanocavity, namely, the cavity formed in the nanometer-scale gap between two nAg particles. Such a nanocavity has been shown to massively enhance the PL emission and decay rate[3,22]. That is why the PL emission of QDs placed in proximity to a single nAg particle and inside a nAg dimer nanocavity was simulated (see the Supplementary Information for more details). The PL emission of a QD is modeled as a two-step process involving excitation and decay. Here, we assume that the excitation rate of the QD is much slower than its total decay rate, so the excited QD decays to its ground state before the next excitation event. In this weak excitation regime, the PL emission enhancement $\tilde{\gamma}_{ems}$ is given by

$$\tilde{\gamma}_{ems} = \tilde{\gamma}_{exc} \frac{\eta}{\eta_0} \qquad (1)$$

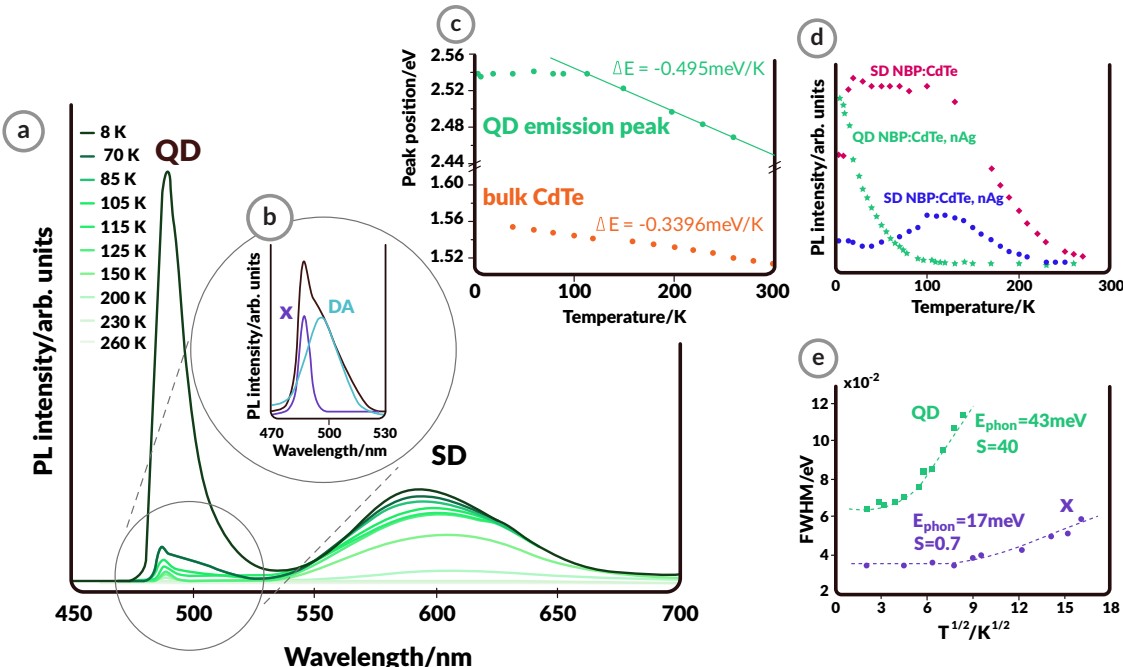

**Fig. 3 | Photoluminescence (PL) temperature dependence of the glass-QDs-nAg (where QDs: quantum dots, nAg: silver nanoparticles) nanocomposite.**
**a** Temperature-dependent PL spectra of NBP:CdTe,nAg (NBP: sodium borophosphate glass). The intensity of the emission strongly increases at low temperatures due to lower phonon-exciton interactions. **b** Excitonic (X) and donor-acceptor (DA) contributions to the QD PL. **c** Comparison of the temperature dependence of the CdTe QD (green) and bulk (orange) peak positions, confirming the quantum dot origin of the QD emission. Linear fit: y = − 0.495*10³*x + 2.599, R² = 0.998. **d** Comparison of the QD and surface defect (SD) PL intensities in NBP:CdTe,nAg, and NBP:CdTe. **e** Full width at half maximum FWHM values of the QD peak and X contribution fitted to the CC model, indicating the weak/strong interaction between the CdTe QD excitons/DA contribution to the PL and lattice phonons.

where $\widetilde{\gamma}_{\mathrm{exc}}$ is the excitation enhancement and $\eta_0 \approx 0.25$ is the quantum yield of the QD in vacuum[54]. The quantum yield $\eta$ of the QD in an optical environment is defined as

$$\eta = \frac{\widetilde{\gamma}_{\mathrm{rad}}}{\widetilde{\gamma}_{\mathrm{rad}} + \widetilde{\gamma}_{\mathrm{nr}} + (1-\eta_0)/\eta_0} \qquad (2)$$

where $\widetilde{\gamma}_{\mathrm{rad}}$ is the radiative decay enhancement and $\widetilde{\gamma}_{\mathrm{nr}}$ is the non-radiative decay enhancement. The total decay enhancement $\widetilde{\gamma}_{\mathrm{tot}} = \widetilde{\gamma}_{\mathrm{rad}} + \widetilde{\gamma}_{\mathrm{nr}}$ determines the increase in the PL lifetime of the QD and is equal to the Purcell factor.

Figures 3e and 3f show numerical calculation results for the PL emission enhancement $\widetilde{\gamma}_{\mathrm{ems}}$ and total decay enhancement $\widetilde{\gamma}_{\mathrm{tot}}$ (the Purcell factor), respectively, for scenarios involving (1) a QD located at a varying distance $z$ in close proximity to a (single) nAg particle (light blue curves in the figures) and (2) a QD embedded in the gap between two nAg particles (nAg dimer) with a gap size of $2z$ (dark green curves in the figures). In both cases, the PL emission enhancement is evaluated for an excitation wavelength of 477 nm and emission wavelength of 513 nm, which correspond to the absorption and emission wavelengths of the CdTe QDs, respectively. Figure 2e reveals that the PL emission enhancement $\widetilde{\gamma}_{\mathrm{ems}}$ is less than 5 for a single nAg particle, but almost 70,000 in the case of a nAg dimer nanocavity. This result implies that emission from the QDs located inside plasmonic nanocavities tends to dominate the PL emission observed in experiments. Meanwhile, the total decay enhancement $\widetilde{\gamma}_{\mathrm{tot}}$ increases to almost 500 and 10,000 as the QD approaches the nAg particle at a distance $z$ of 1 nm (and a gap of 2 nm for the nAg dimer), as shown in Fig. 2f. The 20-fold increase in $\widetilde{\gamma}_{\mathrm{tot}}$ can be accounted for by (i) the extreme field confinement property of the plasmonic nanocavity and (ii) the fact that the first-order plasmonic resonance of the nanocavity is spectrally similar to the emission frequency of the CdTe QD, as demonstrated in the Supplementary Information. It is also important to emphasize that

the numerical results only provide optimistic estimates of the PL intensity and decay enhancements since QDs are unlikely to be optimally located 1 nm from the nAg particle. Hence, the numerical simulations strongly suggest that the plasmonic environment provided by a single nAg particle cannot account for the increase in both the luminescence intensity and decay lifetime; therefore, the observed experimental results are most likely caused by the plasmonic dimer nanocavity environment.

To determine the processes that are responsible for the light emission, temperature-dependent studies were carried out, where the short-wavelength PL band revealed a multicomponent character. The solid-state nature of these structures allows their optical properties to be studied at low temperatures (down to 8 K), which cannot be achieved with water-dispersed plasmonically enhanced luminescent systems. At low temperatures, the total emission intensity of the NBP:CdTe,nAg glass gradually increases (Fig. 3a). The multicomponent character can be split into exciton (X) and donor-acceptor (D-A) contributions (Fig. 3b). The spectra change drastically with temperature, showing differences in the band positions (Fig. 3c) and intensities (Fig. 3d). While the position of the narrow component of the QD band remains constant below 80 K, further heating leads to a decrease in the peak energy (green points in Fig. 3c). This effect has been observed for excitonic transitions in low-dimensional CdTe structures and is mainly attributed to the dependence of the energy gap on temperature[55]. The slope of the dependence of the QD emission peak position on temperature (thermal coefficient, ΔE) is 0.495 meV/K, which is larger than the value for wide CdTe quantum wells and bulk CdTe (0.3396 meV/K)[56]. As the thermal coefficient increases with the decrease in the dimensionality of the system[55], it can be concluded that the narrow PL corresponds to exciton recombination in the CdTe QDs of NBP:CdTe,nAg. The dependence of QD and SD peak intensities on temperature varies between NBP:CdTe and NBP:CdTe,nAg. In low-temperature measurements, the intensity of the SD emission increases

with decreasing temperature down to 100 K and stays constant below in the nonplasmonic glass. However, the same band behaves differently in the plasmonic glass. There, the SD band's intensity grows with decreasing temperature down to 100 K, while at even lower temperatures it diminishes. At the same time, the QD emission significantly increases at temperatures below 100 K. We suggest that this behavior is caused by the preferential energy transfer to QD states, facilitated by plasmonic enhancement, which hinders the SD states from being effectively excited.

Additionally, closer inspection of the temperature behavior of the X component and QD total peak intensities reveals distinct behavior of the active centers related to the respective PL features. Their FWHM values reflect the configurational coordinate model (CC) that describes electron-phonon coupling according to the formula[57,58]

$$FWHM = [8\ln(2)S]^{1/2}E_{phon}\left[\coth\left(E_{phon}/2kT\right)\right]^{1/2} \quad (3)$$

where k is the Boltzmann constant; T is the temperature; $E_{phon}$ is the energy of the vibration mode of the excited state; and S is the Huang–Rhys factor. The CC model results are plotted in Fig. 3e. For the X line, S = 0.7, while the QD FWHM results are best fitted by S = 40. A Huang-Rhys factor S > 1 suggests the presence of strong electron-lattice coupling, in this case with phonons having an energy of $E_{phon}$ = 43 meV. This is a much higher value than the typical phonon energy in CdTe (21.7 meV[59]), which means that the strength of the coupling leads to lattice distortion and a local decrease in the lattice constant.

In order to understand the origin of the enhanced QD emission upon interaction with plasmonic NPs in the optically active plasmonic glass, the power-dependent PL response was investigated. It suggests that coherent amplification occurs in the whole material which can be envisioned as an ensemble of active QD-populated plasmonic nanocavities (Fig. 4a). For the spectra excited at powers up to 100 mW, the intensities of the emitted light increase linearly. Above this threshold, the PL intensities fit a different, much steeper linear trend (Fig. 4b). Additionally, over 165 mW, the system becomes saturated and a plateau is observed. This shows that, under particular excitation conditions (in our case over 100 mW), the system exhibits coherent light amplification. The emergence of plasmon-driven lasing is corroborated by the fact that, in particular areas of the material sample where the density is particularly large, a dip in the extinction is observed (red

arrow, Fig. 4c). This evident feature coincides with the narrow PL band of the material. Such a decrease in absorption is characteristic of systems where coupling between a QD and plasmonic microcavity occurs[60,61]. Judging by the observed intensity of the effect, QDs in a plasmonic nanocavity are in the intermediate coupling regime, showed by Gray and his group[61]. It is worth noting that such an extinction dip appears in optically-pumped plasmonic metamaterials, where coherent light amplification is obtained by accomplishing the dominance of radiative coupling over internal losses[62–64]. This demonstrates the potential of these glass-based plasmonic composites as highly efficient luminescent glasses, indicating that further development of the system would be advantageous in future studies.

## Discussion

Exciton-plasmon interactions provide new possibilities such as amplification of emission and faster time decays, which have been studied in various plexcitonic systems. However, most real-life applications require a hitherto unavailable solid material ready to be incorporated in integrated devices. In this work, we have obtained and demonstrated a material which appears to be the only solid volumetric QD-based plexcitonic material exhibiting increased spontaneous emission, three orders of magnitude decreased lifetime, and narrow-band emission. Summarizing the optical parameters achieved in our system resulting from the QDs situated in a plasmonic nanocavity, we find that: (a) the demonstrated FWHM of the QD exciton emission is the lowest (13 nm in RT) compared with other reported systems (~ 20 nm by Wen et al.[65], ~ 20–30 nm by Hoang et al.[2], ~ 50 nm by Bao et al.[66]); on the other hand (b) exciton PL lifetimes achieved in our material ($\tau$ = 60–90 ps) are of the same order of magnitude as reported previously (~ 11 ps by Hoang et al.[2], ~ 83 ps Bao et al.[66], ~ 700 ps by Akselrod et al.[67]). Previously reported related systems include: (a) surface-based structures with the emitting medium located in a plasmonic gap between a continuous metallic film and metallic nanoparticles placed on top of the emitter[2,65–67], (b) colloidal plasmonic-excitonic assemblies, which can be considered analogous to our glasses[68–70]; as well as (c) other cavity-based systems, such as photonic crystal-coupled emission (PCCE), where a photonic crystal is coupled with an optical cavity[65,71,72].

By fabricating volumetric nanocomposites simultaneously doped with QDs and plasmonic nanoparticles, this study has demonstrated the production of bulk materials with ultranarrow and ultrafast spontaneous emission under continuous-wave laser excitation at room temperature. A significant decrease in the emission lifetime paves the

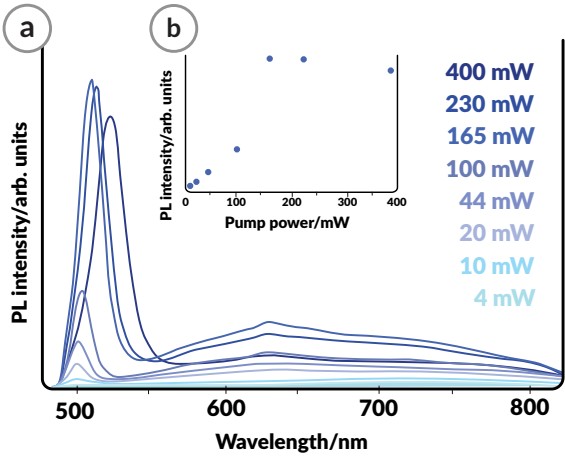

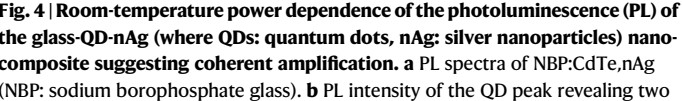

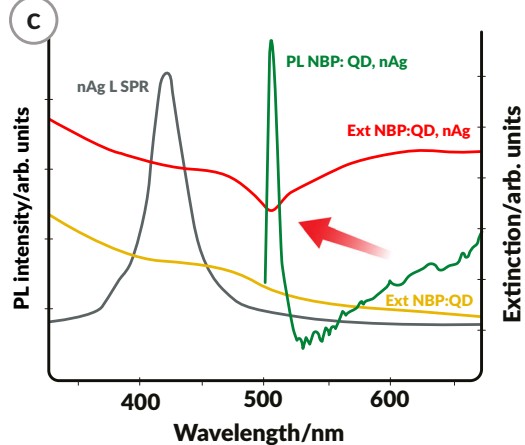

**Fig. 4 | Room-temperature power dependence of the photoluminescence (PL) of the glass-QD-nAg (where QDs: quantum dots, nAg: silver nanoparticles) nanocomposite suggesting coherent amplification. a** PL spectra of NBP:CdTe,nAg (NBP: sodium borophosphate glass). **b** PL intensity of the QD peak revealing two linear trends and a saturation of the emission ($\lambda_{em}$ = 405 nm). **c** Representative extinction of NBP:QD, Ag (red line) collected from a selected area in the sample compared to the PL of the material (green line), extinction of NBP:QD (orange line) and LSPR of nAg (gray line).

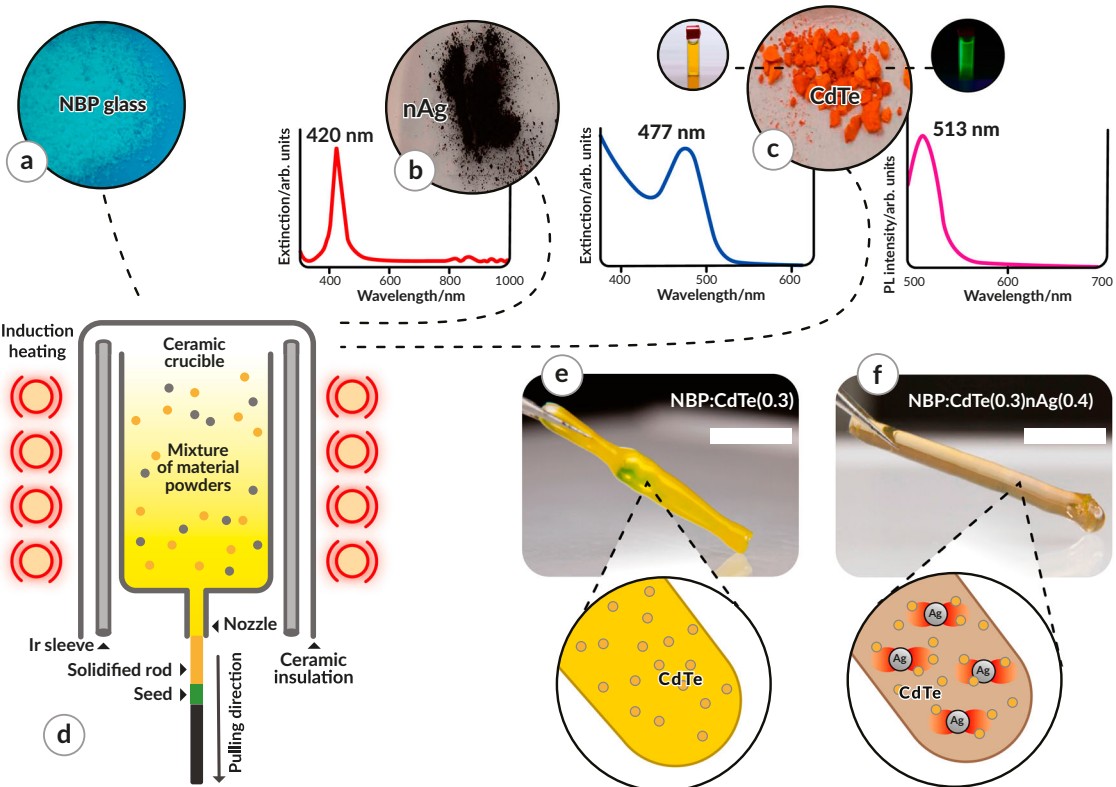

**Fig. 5 | Fabrication of the bulk glass-QDs-nAg (where QDs: quantum dots, nAg: silver nanoparticles) nanocomposite with the NanoParticle Direct Doping (NPDD) method.** Raw materials: (**a**) sodium borophosphate (NBP) glass powder, (**b**) nAg powder and the extinction spectrum of nAg-doped NBP glass, and (**c**) CdTe QD powder, including its aqueous dispersion with extinction (left) and PL (right) spectra. **d** Scheme of the NPDD method. Images and schemes of the fabricated NBP glass rods doped with (**e**) 0.3 wt% CdTe quantum dots, NBP:CdTe(0.3) and (**f**) 0.3 wt% CdTe QDs and 0.4 wt% nAg, NBP:CdTe(0.3)nAg(0.4). CdTe QD diameter = 1.5 nm, $\lambda_{em}$ = 510 nm, and nAg diameter = 20 nm. Scale bars in Figs. 5e and 5f show the length of 2 cm.

way for the application of such materials in high-speed optical devices in fields where high repetition rates are required, e.g., telecommunication transmitters/receivers/transceivers, which is a field that has been vastly inaccessible for QD-based systems due to the relatively long radiative lifetimes of QDs[2]. Moreover, the ultranarrow PL can enable the utilization of light sources with precisely-defined emission wavelengths, such as screen pixels. Importantly, these properties are present at room temperature, in contrast to other systems described thus far in which sharp PL (from QDs) is achieved mostly at low temperatures. Obtaining an even narrower emission could be achieved by further engineering of the system, e.g., coupling with whispering gallery mode resonators, whereby a resonator works as a cavity enabling lasing action[73]. The strong coupling suggested by the results and short PL decay times indicate that single-photon light emission with a high repetition rate is possible with the described composite. With an adequate emitter confined in a cavity, it becomes possible to control, for instance, the spin interactions in a magnetic field, which are relevant in spintronics. Such materials are potentially applicable to a new generation of transistors or storage devices, such as quantum computers[74].

All of the above results, combined with the facile fabrication enabled by the NPDD method, demonstrate the rapid and scalable fabrication of active plasmonic systems, which has not previously been achieved. Moreover, straightforward doping allows for the selection of adequate optically active elements, leading to tailored optical properties. The properties can be further modified by introducing different kinds of NPs with various geometries, including spherical and anisotropic NPs with several LSPR bands (nanowires, nanospheres, etc.); various compositions, including plasmonic NPs made of precious metals/semiconductors to tune the LSPR; and various configurations, including single doping and co-doping with several emitters[41] or with several kinds of plasmonic NPs.

Furthermore, the optical properties of NPDD-grown materials could be tailored by combining various optically active and plasmonic elements that have different sizes, geometries, compositions, or configurations (single doping or co-doping). This flexibility results in their potential application in efficient multicolor light sources, display devices, light amplifiers, telecommunications, etc., where different working ranges are required. Mechanical treatment, which leads to rod fragmentation, also results in changes in the form and shape of the composite; thus, this capability supports the generation of a wide range of microbiosensors for the detection of biomolecules such as proteins, antibodies, and biomarkers. While combining QD functionality with plasmonic nanomaterials has already brought many cutting-edge applications, starting from low-threshold lasers, through quantum information processing, to biodevices, the ever accelerating progress of machine learning and artificial intelligence should facilitate swift advancement of plexcitonic devices for integration in real life applications.

## Methods

The matrix for the composites, $Na_5B_2P_3O_{13}$ glass (NBP), was prepared from $Na_2CO_3$ (99.99%), $NH_4H_2PO_4$ (analytically pure), and $H_3BO_3$ (99.99%) powders (Fig. 5a). These powders were mixed in a 5:6:4 molar ratio in an alumina mortar and heated to 920 °C. After solidifying at room temperature, the obtained glass was mechanically ground in a mortar into a powder.

All the composites were fabricated using NPDD[39] which uses the principles of the micro-pulling down (μ-PD) method[75]. The raw material for the fabrication of 0.3 wt% CdTe QDs and 0.4 wt% nAg, written as

NBP:CdTe(0.3)nAg(0.4) throughout the paper, was analogously prepared. 0.4 wt% nAg was added (20–30 nm, SkySpring Nanomaterials, Fig. 5b) to the powdered NBP. Commercially available CdTe QDs ($\lambda_{max}$ = 510 nm, diameter 1.5 nm, PlasmaChem) were also directly added to the powdered NBP (representing 0.3 wt% of the material, Fig. 5c). Characterization of the nanomaterials is shown in Supplementary Fig. 2; transmission electron microscopy (TEM) images and powder X-ray diffractometry (XRD) results with Rietveld analysis confirmed that the average size of nAg was ca. 25 nm. While the XRD diffractometer for CdTe QDs confirmed that their size is below 10 nm, TEM images show small QDs of the size ca. 1.5 nm. To fabricate the quantum dot-doped NBP:CdTe(0.3) glass nanocomposite, only QDs were utilized.

All the materials were mechanically mixed and placed in an alumina crucible, which was inductively heated by an iridium tube using a commercial μ-PD apparatus (Cyberstar). The heating was adjusted so that liquid material could be visible at the end of the crucible shaper. Convection provided additional mixing of the melt. Nanocomposites in the form of rods were then pulled down at a rate of 1–2 mm/min in a nitrogen atmosphere. The scheme of the NPDD thermal setup that was used for the production of the materials is illustrated in Fig. 5d.

PL spectra were obtained by exciting the nanocomposites with either a 405 nm laser diode or a 488 nm continuous-wave Ar$^+$ laser and recorded with a Jobin Yvon HR460 monochromator equipped with a 1200 L/mm grating and a lock-in technique. The PL signals were collected by cooled photomultipliers (EMI 9658 BM and Hamamatsu 5509-72). Temperature-dependent measurements were performed with a cryostat working in a closed cycle cooling system. The PL lifetime measurements were performed using a Hamamatsu streak camera with an OPerA-Solo optical parametric amplifier and a Coherent Libra femtosecond laser (1 mJ, 89 fs) tuned to 450 nm with PL collection at 480 nm or 625 nm.

## Data availability

Source data of the results presented in the Figures are available in the repository under the https://doi.org/10.6084/m9.figshare.25288579[76].

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

## Acknowledgements

This paper is dedicated to the memory of our dear co-worker and friend Hańcza Barbara Surma, who passed away while this manuscript was being completed. We want to thank Prof. Stephen K. Gray from the Center for Nanoscale Materials, Argonne National Laboratory, Lemont,

Illinois, USA, for support in data interpretation. P.P., R.N., D.A.P thank the TEAM/2016-3/29 grant within the TEAM program of the Foundation for Polish Science co-financed by the European Union under the European Regional Development Fund. P.P., A.S., D.A.P. thank ENSEMBLE[3] Project (MAB/2020/14) which is carried out within the International Research Agendas Program (IRAP) of the Foundation for Polish Science co-financed by the European Union under the European Regional Development Fund and Teaming Horizon 2020 program of the European Commission. M.B., A.S., P.O., M.G., D.A.P. thank the MAESTRO project "New Generation Plasmonic Materials" financed by the National Science Center of Poland (G.A. 2011/02/A/ST5/00471) for support of this work. O.H. acknowledges financial support from the Science Foundation Ireland (SFI) via grant number 18/RP/6236.

## Author contributions

D.A.P. provided the financial support, D.A.P. and M.G. conceived the idea, M.B., R.N., H.B.S., P.O., M.G. and A.S. performed the experiments, P.P., N.K., H.B.S., A.S., O.H. and D.A.P. analyzed and interpreted the data, P.P., N.K., O.H. and D.A.P. wrote the paper, W.R.R., O.H. and D.A.P. supervised the project, all authors revised the paper.

## Competing interests

M.G. and D.A.P. declare the following competing interests: Polish patent by ENSEMBLE[3] sp. z o.o., inventors: Andrzej Kłos, Marcin Gajc, Katarzyna Sadecka, Dorota Anna Pawlak, patent in force, number PAT.219519, including the description of fabrication technology of glass rods doped with metallic and/or semiconductor nanoparticles; European patent by Institute of Electronic Materials Technology, inventors: Andrzej Kłos, Marcin Gajc, Katarzyna Sadecka, Dorota Anna Pawlak, patent in force, number EP2570396, including the description of fabrication technology of glass rods doped with metallic and/or semiconductor nanoparticles. The remaining authors declare no other competing interests.
