## [Peer Review File · Nature Communications]

Ultrafast Photoluminescence and Multiscale Light Amplification in Nanoplasmonic Cavity GlassREVIEWER COMMENTS

Reviewer #1 (Remarks to the Author):

The authors have prepared a tape of bulk plasmon-exciton composites material, which achieves ultranarrow and ultrafast amplified photoluminescence. They also explained the key role of the QD-populated plasmonic nanocavities through FDTD simulations. These results provide reference for the follow-up research of bulk plasmon-exciton composites material. However, there are still some places need revised in the article.

1. The mechanism and reason about luminescence enhancement and lifetime reduction of QDs emitter in nanoplasmonic environment should be given, combined with the FDTD simulation results and interaction between QDs and plasmonic field.

2. The sentences “In the nonplasmonic NBP:CdTe, the intensity of the SD band increases as the temperature decreases to 100 K; below 100 K, saturation is observed. However, in the NBP:CdTe,nAg glass, the intensity of this band gradually decreases below 100 K in favour of QD emission.” on Page 13 needs further explanation. What is the effect of nanoplasmonic field on SD PL intensity under different temperature?

3. In figure 5(b), there are only two PL intensity points between 100mW and 165mW, which should be added to better illustrate the linear relationship. And please make clear the reason for the plateau of emission intensity when the excited power over 165 mW.

4. In Fig. S1, why did the plasmon resonance peak of nAg disappear in the absorbance spectrum? Did the nAg agglomerate during the process of bulk

glass-QD-nAg nanocomposite?

Reviewer #2 (Remarks to the Author):

Piotrowski et al., demonstrate an interesting experimental work where the volumetric glass-based plexcitonic nanocomposites is simultaneously doped with cadmium telluride quantum dots (CdTe QDs) and plasmonic silver nanoparticles (nAg) via the NPDD method, exhibiting ultrafast and spectrally ultranarrow luminescence. The distribution of QDs within the material is demonstrated with fluorescence lifetime imaging microscopy. Finite difference time-domain (FDTD) simulations has been carried out by the authors to validate the exceptional optical response originating from the interaction between CdTe quantum emitters and the self-arranged framework of plasmonic NPs in the obtained materials. Moreover, the power dependence of photoluminescence, PL, indicates coherent light amplification in the system, while temperature dependence provides additional insight into the nature of the emission. The results presented are very impressive and definitely of immense value to the broad audience of semiconductor industry, photo-plasmonics and biochemical sensor technologies. This interesting work may be considered for publication, provided the authors address the below mentioned comments.

1. The introduction section is not comprehensive. The authors should provide the relevant background from the perspective of key high highlights of the research.
2. The motivation of this work is not clearly mentioned in the introduction.
3. The reviewer understands that the results presented in this work has tremendous applications in myriad fields of science and technology. It is important to dedicate a section towards the end of the manuscript, with a title '*futuristic scope and perspectives*' presenting the same elaborately.
4. The performance of the presented device framework should be compared and contrasted with the ones that out there in the literature (articles and patents).
5. Figure 1 is extremely confusing. The inset should be presented separately as a separate figure as the same scale bar does not hold good in the case of the inset. The authors should pay close attention to such aspects.
6. The English language and prose should be improved throughout the manuscript. The quality of figures also needs improvement.
7. What are the components of the glass material that might show possible interference and background signals? How did the authors overcome this effect? What are the steps considered to avoid such experimental artifacts?
8. How does the lifetime change in terms of mono-bi-tri-exponentiality with the addition of plasmonic Ag.
9. Authors should comment on the radiative and non-radiative decay components of the systems before and after incorporating plasmonic Ag. How would be effects change for differently sized and shaped AgNPs.
10. It is instructive to include the discussion on metal-dielectric hybrid coupling effects such as those observed in related technologies, for instance photonic crystal-coupled emission (PCCE).
11. The experimental section does not provide complete information. It does not have any references. Did the authors develop the techniques for the first time? If not, please provide appropriate references. The modifications in the conventional experiments should be explicitly mentioned.
12. The authors should briefly discuss the meaning and pertinence of plexcitonic material. Further, the abbreviation must be defined upon its first occurrence in the manuscript and should not be repeated.

13. How did the authors overcome the effect of defect state in the hybrid mixture? What are the demerits/limitations of the proposed technology.
14. All the nanomaterials presented in this work should be characterized using TEM and XRD techniques. Presenting only the extinction profiles would not suffice, even if the materials are procured from company.
15. The authors state 'On the other hand, it is likely that randomly distributed CdTe QDs could be located inside an optical nanocavity, namely, the cavity formed in the nanometer-scale gap between two nAg particles'. It is important support such claims using appropriate literature from crysores colloids/nano-assemblies and related technologies.
16. How did the authors overcome the surface-induced damping and the effect of Ohmic losses in hindering the luminescence intensity? Were there any spacer nanolayer considered around the plasmonic AgNPs?
17. The temperature dependent explorations should be performed by both increasing and decreasing the temperature. The related discussion is missing.
18. The conclusions section should be presented where the authors should clearly mention the key highlights of the work.
19. From the plasmonics perspective, why did the authors choose to work with this particular quantum dot and plasmonic Ag, while there are other metallic systems that can be explored in the same wavelength range.

In summary, this very interesting work may be considered for publication, provided the authors address the above mentioned comments.

We thank the reviewers for their positive evaluation of our manuscript. We highly appreciate their valuable comments that will help us improve the quality of the article. Below, we attach responses to the reviewers' remarks. All the changes in the manuscript are marked in yellow.

Reviewer #1

The authors have prepared a tape of bulk plasmon-exciton composites material, which achieves ultranarrow and ultrafast amplified photoluminescence. They also explained the key role of the QD-populated plasmonic nanocavities through FDTD simulations. These results provide reference for the follow-up research of bulk plasmon-exciton composites material. However, there are still some places need revised in the article.

1. The mechanism and reason about luminescence enhancement and lifetime reduction of QDs emitter in nanoplasmonic environment should be given, combined with the FDTD simulation results and interaction between QDs and plasmonic field.

We thank the reviewer for the comment and suggestion. It is well known that an emitter placed in the direct vicinity of a solitary metallic nanoparticle experiences quenching caused by enhancement of the nonradiative decay. Typically, this effect is observed for emitters interacting with a metal nanostructure within a distance closer than 10 nm. However, it has been shown (ACS Photonics 2018, 5, 186-191) that 'quenching can be quenched' if the emitter is situated in a nanoplasmonic cavity. This way, the radiative decay rate is larger than the non-radiative decay rate and the re-emission of its energy is promoted (ACS Photonics 2018, 5, 186-191). These observations are in accordance with our findings. Our FDTD calculations compare the optical parameters of two systems: a CdTe QD interacting with one or two Ag nanoparticles. They demonstrate that, indeed, the enhancement of the optical properties observed in our experiments can only originate from a nanosystem where a CdTe QD is placed between at least two Ag nanoparticles that form a plasmonic nanocavity (Figures 3e and 3f, Finite-Difference Time-Domain Simulations paragraph in the Supplementary Material).

2. The sentences "In the nonplasmonic NBP:CdTe, the intensity of the SD band increases as the temperature decreases to 100 K; below 100 K, saturation is observed. However, in the NBP:CdTe,nAg glass, the intensity of this band gradually decreases below 100 K in favour of QD emission." on Page 13 needs further explanation. What is the effect of nanoplasmonic field on SD PL intensity under different temperature?

We appreciate the reviewer mentioning this interesting observation. In low temperature measurements, the intensity of the SD emission increases with decreasing temperature down to 100 K and stays constant below this in the nonplasmonic glass. However, the same band behaves differently in the plasmonic glass. There, the SD band's intensity grows with decreasing temperature down to 100 K, while in even lower temperatures it diminishes. At the same time, the QD emission significantly increases at temperatures below 100 K. We suggest that this behaviour is caused by the preferential energy transfer to QD states, facilitated by plasmonic enhancement, which hinders the SD states from being effectively excited. The text has been modified accordingly.

3. In figure 5(b), there are only two PL intensity points between 100mW and 165mW, which should be added to better illustrate the linear relationship. And please make clear the reason for the plateau of emission intensity when the excited power over 165 mW.

We thank the reviewer for this suggestion. Above 165 mW, heat-induced effects are expected due to the high power of the excitation. Insufficient cooling can cause e.g. sample damage on the nanoscale which hinders further increase of the signal and leads to the plateau of emission intensity. We also follow the reviewer's suggestion to remove the linear trends which previously had been drawn as a guide to the eye. Figure 5b has been modified accordingly.

4. In Fig. S1, why did the plasmon resonance peak of nAg disappear in the absorbance spectrum? Did the nAg agglomerate during the process of bulk glass-QD-nAg nanocomposite?

In Figure S1, the absorbance band of NBP:CdTe,nAg is broader than the band observed for NBP:CdTe. We interpret this broadening as the overlap of the CdTe QDs absorption band and the nAg LSPR band, which should be expected in the range of 400-450 nm. Our previous observations indicate that the agglomeration of nanoparticles in the glass is hampered by microcavitation in the capillary of the crucible during fabrication of the material.

Reviewer #2

Piotrowski et al., demonstrate an interesting experimental work where the volumetric glassbased plexcitonic nanocomposites is simultaneously doped with cadmium telluride quantum dots (CdTe QDs) and plasmonic silver nanoparticles (nAg) via the NPDD method, exhibiting ultrafast and spectrally ultranarrow luminescence. The distribution of QDs within the material is demonstrated with fluorescence lifetime imaging microscopy. Finite difference time-domain (FDTD) simulations has been carried out by the authors to validate the exceptional optical response originating from the interaction between CdTe quantum emitters and the self-arranged framework of plasmonic NPs in the obtained materials. Moreover, the power dependence of photoluminescence, PL, indicates coherent light amplification in the system, while temperature dependence provides additional insight into the nature of the emission. The results presented are very impressive and definitely of immense value to the broad audience of semiconductor industry, photo-plasmonics and biochemical sensor technologies. This interesting work may be considered for publication, provided the authors address the below mentioned comments.

1. The introduction section is not comprehensive. The authors should provide the relevant background from the perspective of key high highlights of the research.

We appreciate the comment and have introduced changes in the introduction following the comments of the reviewer to improve its reception. The new text is marked in yellow on pages 2-5.

2. The motivation of this work is not clearly mentioned in the introduction.

The main motivation for this work was to demonstrate the possibility of obtaining a real material containing both excitonic and plasmonic species that exhibits plasmon-exciton effects in the volume. Up to now, this kind of phenomena was predominantly shown in 2D architectures. This motivation has been included in the amended version of the introduction.

3. The reviewer understands that the results presented in this work has tremendous applications in myriad fields of science and technology. It is important to dedicate a section towards the end of the manuscript, with a title ‘futuristic scope and perspectives’ presenting the same elaborately.

We are grateful for such a positive feedback regarding our work. Following the reviewer’s advice, the Outlook section has been renamed and extended. We initially changed the title of the last section to the one suggested by the reviewer. However, following the comment no. 18 about including the summary of the conclusions there, we would need to rename the last chapter to “Conclusions, futuristic scope and perspectives”, which appears to be too long. As a result, we propose a new title of the section: “Conclusions and perspectives”.

4. The performance of the presented device framework should be compared and contrasted with the ones that out there in the literature (articles and patents).

Following the comment above, the comparison with the literature has been added to the “Conclusions and perspectives” section.

5. Figure 1 is extremely confusing. The inset should be presented separately as a separate figure as the same scale bar does not hold good in the case of the inset. The authors should pay close attention to such aspects.

We thank the reviewer for pointing out this issue. In the amended version of the manuscript, we have provided a new version of Figure 1 to avoid misunderstanding.

6. The English language and prose should be improved throughout the manuscript. The quality of figures also needs improvement.

Following this comment, we have paid particular attention to the language and the quality of figures in the resubmission.

7. What are the components of the glass material that might show possible interference and background signals? How did the authors overcome this effect? What are the steps considered to avoid such experimental artifacts?

The glass is transparent in the visible range where the emission is observed. Therefore, we do not expect any direct background signal from the matrix. Commercially available nanoparticles were selected from different types by taking into consideration any potential interference and choosing those which did not introduce any additional signal. Finally, the growth processes were carried out in well-controlled conditions with system components dedicated to glass fabrication, thoroughly cleaned after each process to avoid contamination from fabrication of other glass materials and of materials of different nature.

8. How does the lifetime change in terms of mono-bi-tri-exponentiality with the addition of plasmonic Ag.

The lifetimes for fitting mono-, bi-, and tri-exponential decay curves are given in Table 1. The PL lifetime decay for the plasmonic material NBP:CdTe,nAg can be fitted with two or three exponential functions, where all decay times are in the range of tens or hundreds of picoseconds. Attempts to fit the decay curves with more than one exponential term for the non-plasmonic materials results in generating decay times of: 1) the same value, indicating that the additional exponential terms are not necessary, or 2) high errors, indicating that the performed fittings are not adequate. Therefore, we decided to keep the values of single exponential decays in the manuscript so that the results are easy to compare between different materials.

Table 1. Lifetimes in ns for mono-, bi-, and triexponential fitting curves (Exp1, Exp2, and Exp3, respectively).

Material		Exp1	Exp2	Exp3
NBP:CdTe,nAg	$t1$	0.0914 ± 0.0098	0.0717 ± 0.0014	0.0551 ± 0.0111
	$t2$		0.1860 ± 0.0057	0.1097 ± 0.0408
	$t3$			0.2424 ± 0.0942
	Adj. R^2	0.98798	0.99952	0.99962
NBP:CdTe	$t1$	2.5198 ± 0.1951	2.5197 ± 49371.4402	1.9176 ± 242.5400
	$t2$		2.5199 ± 35936.9406	1.9181 ± 621.4131
	$t3$			10.2191 ± 0.4504
	Adj. R^2	0.99819	0.99783	0.99939
QDs in H ₂ O	$t1$	29.1579 ± 0.2125	16.2120 ± 0.7116	16.2121 ± 2.3836
	$t2$		42.9110 ± 1.5232	42.9082 ± 43144.7324
	$t3$			$42.9143 \pm 46886,6579$
	Adj. R^2	0.99805	0.99975	0.99975

9. Authors should comment on the radiative and non-radiative decay components of the systems before and after incorporating plasmonic Ag. How would be effects change for differently sized and shaped AgNPs.

The particular shape of a plasmonic nanoparticle does, indeed, have an influence on the specific value of the radiative and non-radiative decay rates of quantum emitters embedded in nanoplasmonic cavities that are made up, for example, from two plasmonic nanoparticles. This influence is discussed in detail in [EPJ Appl. Metamat. 5, 6 (2018)] where it is demonstrated that increasing the NP's facet width, the field enhancement inside the nanocavity is weakened in exchange for a larger effective volume. The analysis in [EPJ Appl. Metamat. 5, 6 (2018)] shows that, in the weak coupling regime, a dimer nanocavity retains its ability to suppress fluorescence quenching and enhance radiative emission of an emitter for facet radius <10 nm. In the strong-coupling regime [EPJ Appl. Metamat. 5, 6 (2018)] shows that a nanocavity with a smaller facet provides a larger field enhancement and couples more strongly with a small number of emitters placed at or close to the center of the nanocavity. On the other hand, a nanocavity with a larger facet can accommodate more emitters and couple more strongly with a large ensemble of emitters.

Regarding the principle incorporation of plasmonic Ag and its influence on radiative and non-radiative decay, we refer to the discussion in point 15.

10. It is instructive to include the discussion on metal-dielectric hybrid coupling effects such as those observed in related technologies, for instance photonic crystal-coupled emission (PCCE).

We thank the reviewer for drawing our attention to the field of PCCE in the context of our work. This comment on PCCE has been included in the "Conclusion and perspectives" section.

11. The experimental section does not provide complete information. It does not have

any references. Did the authors develop the techniques for the first time? If not, please provide appropriate references. The modifications in the conventional experiments should be explicitly mentioned.

Since the materials were fabricated using the NPDD method developed by us, the experimental section has been amended with this information.

12. The authors should briefly discuss the meaning and pertinence of plexcitonic material. Further, the abbreviation must be defined upon its first occurrence in the manuscript and should not be repeated.

We thank the reviewer for pointing this out. The discussion about plexcitonic materials has been added to the Introduction and other modifications have been introduced accordingly.

13. How did the authors overcome the effect of defect state in the hybrid mixture? What are the demerits/limitations of the proposed technology.

We are positive that the surface defect emission in NBP:CdTe(0.3)nAg(0.4) is hindered by the plasmonic enhancement of the excitonic PL of CdTe QDs. As the LSPR of nAg is tuned more with the excitonic emission wavelength, QD luminescence is promoted in the system and the SD transition of lower energy is not excited efficiently in these conditions.

14. All the nanomaterials presented in this work should be characterized using TEM and XRD techniques. Presenting only the extinction profiles would not suffice, even if the materials are procured from company.

Following the reviewer's comment, the suggested characterization results have been added to Supplementary Materials.

15. The authors state 'On the other hand, it is likely that randomly distributed CdTe QDs could be located inside an optical nanocavity, namely, the cavity formed in the nanometer-scale gap between two nAg particles'. It is important support such claims using appropriate literature from crysoret colloids/nano-assemblies and related technologies.

We thank the reviewer for the comment. Just as discussed in the reply to comment 1 of Reviewer #1, we can say that the statement is backed up by our observations of photoluminescence enhancement in the plasmonic material and supported by the theoretical modelling of the system. It is well known that an emitter placed in the direct vicinity of a solitary metallic nanoparticle experiences quenching caused by enhancement of the nonradiative decay. Typically, this effect is observed for emitters interacting with a metal nanostructure at a distance closer than 10 nm. However, it has been shown (ACS Photonics 2018, 5, 186-191) that 'quenching can be quenched' if the emitter is situated in a nanoplasmonic cavity. This way, the radiative decay rate is larger than the non-radiative decay rate and the re-emission of its energy is promoted (ACS Photonics 2018, 5, 186-191). These observations are in accordance with our findings. The FDTD calculations compare the optical parameters of two systems: a CdTe QD interacting with one or two Ag nanoparticles. They demonstrate that, indeed, the enhancement of the optical properties observed in our experiments can only originate from a nanosystem where a CdTe QD is placed between at

least two Ag nanoparticles that form a plasmonic nanocavity (Figures 3e and 3f, Finite-Difference Time-Domain Simulations paragraph in the Supplementary Material).

16. How did the authors overcome the surface-induced damping and the effect of Ohmic losses in hindering the luminescence intensity? Were there any spacer nanolayer considered around the plasmonic AgNPs?

The luminescence quenching frequently observed in the plasmonic-based nanosystems can be overcome by placing the emitter in the plasmonic nanocavity formed by several metal nanoparticles. In the studied material, the random distribution of QDs and nAg supports the generation of the systems required to observe the reported enhancement. No dedicated spacer nanolayer was applied on either nAg or QDs. However, as both QDs and plasmonic nanoparticles were placed in the glass matrix, the observed exciton emission enhancement suggests that, within the solidification process the glass naturally formed a proper distance between the sufficient number of these two species, enabling the enhancement.

17. The temperature dependent explorations should be performed by both increasing and decreasing the temperature. The related discussion is missing.

We thank the reviewer for the interesting suggestion. While temperature-dependent studies at low temperatures were performed, we are concerned that high-temperature studies cause degradation of the material. Heat-induced effects can already be observed in Figure 5b as the intensity plateau for high excitation powers and the same effects can be expected in high temperature measurements, especially given that NBP is a low-melting point glass.

18. The conclusions section should be presented where the authors should clearly mention the key highlights of the work.

Following the reviewer's comment, the conclusions of the work have been highlighted in the closing section of the manuscript: "Conclusions and perspectives".

19. From the plasmonics perspective, why did the authors choose to work with this particular quantum dot and plasmonic Ag, while there are other metallic systems that can be explored in the same wavelength range. In summary, this very interesting work may be considered for publication, provided the authors address the above mentioned comments.

Ag nanoparticles have many advantages over other plasmonic nanosystems. In particular, they are generally available commercially and/or easily synthesized; they exhibit strong plasmonic properties that were well-studied in the literature; and are relatively stable chemically. On the other hand, gold nanoparticles are much more expensive than silver, which would result in a much higher cost of the experiment where relatively high quantities of the materials are necessary. Additionally, the nAg LSPR peak in NBP glass overlaps with the blue laser excitation which is potentially important for future devices. That being said, studies with other metallic systems are also being carried out currently by our group as a follow-up to this manuscript.

REVIEWERS' COMMENTS

Reviewer #1 (Remarks to the Author):

The authors have significantly improved the manuscript and addressed most of the concerns of the reviewers.

Reviewer #2 (Remarks to the Author):

The authors address the reviewer comments well.